# Influence of Printing Angulation on the Accuracy (Trueness and Precision) of the Position of Implant Analogs in 3D Models: An In Vitro Pilot Study

Noelia García [1], Miguel Gómez-Polo [2,*], Miriam Fernández [1], José Luis Antonaya-Martín [1], Rocío Ortega [3], Cristina Gómez-Polo [4], Marta Revilla-León [5,6,7] and Rocío Cascos [1,2,3]

1   Department of Nursing and Estomatology, Faculty of Health Sciences, Rey Juan Carlos University,
    28922 Madrid, Spain; ngarciahernanz@gmail.com (N.G.); miriam.fernandez@hotmail.com (M.F.);
    joseluis.antonaya@urjc.es (J.L.A.-M.); rcascos@ucm.es or rocio.cascos@universidadeuropea.es (R.C.)
2   Department of Conservative Dentistry and Prosthodontics, Faculty of Dentistry,
    Complutense University of Madrid, 28040 Madrid, Spain
3   Dental Clinic Department, Faculty of Biomedical Sciences, European University of Madrid,
    28670 Madrid, Spain; rocio.ortega@universidadeuropea.es
4   Department of Surgery, Faculty of Medicine, University of Salamanca, 37007 Salamanca, Spain;
    crisgodent@usal.es
5   Department of Restorative Dentistry, School of Dentistry, University of Washington, Seattle, WA 98195, USA;
    marta.revilla.leon@gmail.com
6   Kois Center, Seattle, WA 98109, USA
7   Department of Prosthodontics, School of Dental Medicine, Tufts University, Medford, MA 02155, USA
*   Correspondence: mgomezpo@ucm.es; Tel.: +34-91-394-2029

**Abstract:** Background: With CAD-CAM, dental models are often fabricated by additive manufacturing (AM) methods. Dental models for implant prostheses involve the manual placement of analogs, which could affect their final fit. Therefore, in this experimental in vitro study, the accuracy of the analogs' position in models printed using AM methods was examined by comparing three different printing orientations. Methods: An STL file was obtained by scanning a master model with an implant placed in the maxillary left central incisor position. Three study groups (n = 10) with varying printing orientation parameters were obtained (0, 45, and 90 degrees). They were digitalized with a laboratory scanner and evaluated with an analysis metrology program. Differences between 3D datasets were measured using the RMS for trueness and SD RMS for precision. The data were statistically analyzed using the ANOVA test at a significance level of $p < 0.05$, followed by the Bonferroni post hoc test. Results: The 45-degrees group showed the best results. Regarding trueness, statistically significant differences were found between the 45- and 90-degrees groups ($p < 0.005$). In terms of precision, statistically significant differences appeared between the 45- and 0-degrees groups ($p < 0.011$) and between the 45- and 90-degrees groups (0.003). Conclusions: The printing angulation parameter affects the accuracy of 3D-printed models. Implant models manufactured at 45 degrees of printing angulation are more accurate than those printed at 0 or 90 degrees.

**Keywords:** dental implant analog; variation angulation; digital impression; 3D models; build angle; 3D printing; additive manufacturing

## 1. Introduction

The application of new technologies in the field of dentistry has evolved over the last 20 years thanks to the development of computer-aided design and manufacturing (CAD-CAM), improving the accuracy of diagnoses and treatments, reducing manufacturing times, and allowing for the customization of different therapeutic alternatives [1]. CAD-CAM technology comprises three fundamental phases: data acquisition, digital design, and fabrication. Data acquisition is performed through intraoral digitization or study models,

using techniques such as cone beam computed tomography (CBCT) or intraoral and facial scanners. The data are then processed in CAD software, where the treatment indicated for each patient is designed. Finally, fabrication can follow one of two routes: subtractive or additive, each with its own particularities and optimal applications.

Subtractive manufacturing involves the removal of material from a solid block using computer-controlled cutting tools. Traditionally associated with the machining and milling process, this method benefits from the wide availability of materials and proven efficiency and accuracy in fabricating dental restorations such as crowns, bridges, and implant frameworks. The subtractive technique can employ materials such as ceramics, nanoceramic resin composites, and metals. Its accuracy is remarkably high, although it is associated with higher material waste generation than additive methods. Subtractive equipment also requires maintenance and significant initial investment.

On the other hand, additive manufacturing (AM), commonly known as 3D printing, builds objects layer by layer from a digital file. The American Society for Testing and Materials (ASTM) has described it as "a process of joining materials to fabricate objects from 3D models, usually layer upon layer, in contrast to subtractive manufacturing methodologies" [2]. Three-dimensional printing technology is subdivided into several techniques, including stereolithography (SLA), material jetting (MJP), direct energy deposition (DLP), material extrusion, powder bed fusion (PBF), sheet lamination, and binder jetting [3]. Among them, the most used technologies are SLA and MJP. Each technology has its particularity in terms of the materials it can process, the achievable resolution, and the production speed. Materials commonly used in dental 3D printing include light-curing resins, thermoplastic polymers, and, to a lesser extent, metals and ceramics. 3D printing is particularly advantageous in producing dental models, surgical splints, and prostheses, where geometric complexity or customization is crucial. In addition, this technique significantly reduces material waste and can be faster for certain types of objects [4].

The choice between subtractive and additive manufacturing depends on multiple factors, including the complexity of the restoration, material requirements, and time constraints. While subtractive manufacturing remains the standard for treatments that demand materials with specific mechanical and esthetic properties, additive manufacturing is emerging as a powerful alternative for applications that benefit from its ability to handle complex geometries and advanced customization. The extensive development and integration of CAD-CAM technology in dentistry has enabled various clinical applications, improving the quality and efficiency of treatments. As technology advances, we are likely to see an expansion in the capabilities and applications of both manufacturing techniques, promoting continued innovation in different dental treatments.

In terms of treatments, many restorations can be fabricated employing these technologies, such as onlays, crowns, fixed partial dentures, veneers, implant abutments, full-mouth reconstructions, or orthodontic splints, among others [5]. In addition, these types of dental restorations are more anatomical and faster to produce compared to the traditional method created by laboratory technicians [6,7].

CAD-CAM technology has allowed clinicians and laboratory technicians to change how treatments are planned and prostheses are fabricated [8], favoring and promoting the use of printed models. A dental model is a reproduction of the teeth and surrounding oral tissues obtained through an impression, either digital or analogically. The models are used for patient diagnosis, as well as for prostheses fabrication [9]. The accuracy with which the models reproduce the different oral situations will be decisive in the prosthesis fit in the mouth, influencing the passive fit of the structures [10].

Printed models are widely used in various branches of dentistry, such as prosthodontics, oral and maxillofacial surgery, implantology, orthodontics, endodontics, and periodontology [11]. Traditionally, these models have been fabricated in stone cast, which carries risks associated with the material, such as degradation, fracture potential, storage space, and loss of surface structure [12]. However, to date, most printed models are obtained by additive manufacturing. Additive manufacturing has unique advantages over

traditional and subtractive methods. Among the benefits of 3D-printed models are the space-saving capabilities and the ability to manufacture customized parts with unique geometric characteristics [1,13–15]. The disadvantages are that the supports must be removed, and the employed resins can irritate the oral mucosa by contact or cause damage by inhalation [16,17].

Among the various AM techniques, DLP technology is becoming increasingly popular for producing dental parts [18–20]. The product is produced layer by layer directly from 3D data by exposing consecutive layers of photoactivated liquid monomers to ultraviolet light and curing according to the required final product shape. This type of technology uses a digital micromirror device (DMD) that projects the image of the 3D object to be manufactured onto the surface of the resin, so depending on the geometry of the part to be printed, these DMDs change their orientation [21,22].

Multiple factors must be considered that affect the accuracy of 3D models. Primarily, the quality of the file from which the model is printed must be taken into account; the fidelity of the printed model begins with the quality of the digitization of the patient's dental data. If there are inaccuracies in the digital scan, these can be reflected throughout the printing process, leading to errors in the final product. Additionally, the technician's expertise in performing the 3D printing is significant, as they are responsible for preparing the model in specific design software and selecting and controlling the printing parameters and the conditions under which printing occurs. The ability to choose the appropriate settings and adjust the printing parameters based on experience and knowledge of the material and machine is crucial for the success of the print.

Regarding the 3D printer, the type of technology used, its resolution, as well as the printing parameters, such as layer thickness and print orientation, are crucial to obtaining more accurate models. As for the printing material, it is essential to select a dimensionally stable material with the appropriate mechanical properties since shrinkage during polymerization and degradation over time can affect the final quality of the printed object. Other factors to consider are those related to the post-curing process, where the material's final strength and dimensional stability are achieved. Insufficient curing can leave the material soft and deformable, while excess can lead to brittleness. Finally, cleaning processes to remove unpolymerized material and finishing methods to remove supports can influence the dental model's final texture and dimensional accuracy. In addition to all these factors related to 3D printing, it is essential to consider various environmental conditions, such as temperature and humidity, which can affect the stability of the models, potentially causing problems in layer adhesion or material curing [11,23–26].

One of the main differences between the 3D-printed and the gypsum-poured stones is the design of the analogs. In conventional models, the implant analog is screwed to the impression transfer, and the cast is poured over it. Therefore, the analog is retained in the casting material during the procedure. In 3D-printed models, the implant analog is placed after the model has been manufactured and post-processed. Hence, multiple factors can affect the accuracy of the analog position, such as the analog design, the positioning method, the number of analogs, or the printing orientation. Thus, discrepancies in the analog positions compared to the real position would lead to the final misfit of the prosthesis.

The knowledge about the influence of printing angulation on the accuracy of 3D implant models is scarce. The literature shows studies that analyze the variation of impression angulation in tooth-supported crowns, complete denture bases, printed orthodontic aligners, or surgical splints. However, few studies have analyzed the analog position in 3D models to fabricate implant-supported prostheses. Due to this, the present study aimed to investigate the accuracy (trueness and precision) of the analog position in printed models while varying the printing orientation (0, 45, and 90 degrees). The null hypothesis was that there would be no significant differences in the 3D position of implant analogs by employing several print orientation parameters for 3D-printed manufacturing.

## 2. Material and Methods

The sample was divided into three groups (n = 10) based on the printing angulation: 0, 45, and 90-degrees (G1, G2, and G3).

The reference STL file was obtained from a maxillary model with an internal hex connection implant placed in the left central incisor position. A scan body (ZI SCV 3.7 ref. 52.020, Dess, Barcelona, Spain) (Figure 1) was torqued to the implant at 15N following the manufacturer's recommendations. The model was digitized by a laboratory scanner (Aidite A-IS Pro, Aidite Tech. Co., Qinhuangdao, China).

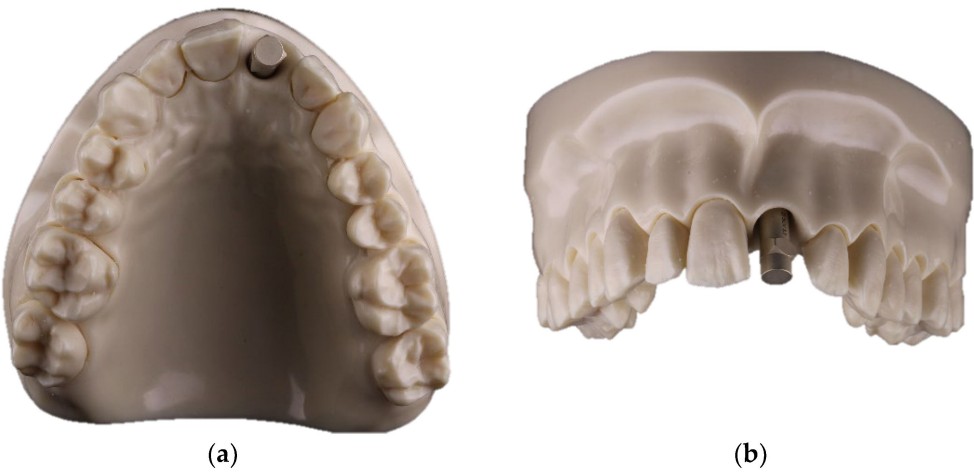

(**a**)          (**b**)

**Figure 1.** Reference model, (**a**) sagittal view, (**b**) frontal view.

The STL was exported to a design software program (Autodesk Meshmixer 3.5, San Francisco, CA, USA) for mesh closure and zoning. Then, it was imported to another CAD software program (DentalCAD 3.0 Galway, Munich, Germany) for the offset setting. A diameter of 0.06 mm was set, corresponding to the diameter of the analog and its radial compensation. The STL file was laminated with a 3D print pre-processing software program (Chitubox®, V.1.9.4 CBD-Tech, Shenzhen, China) (Figure 2) [27–29].

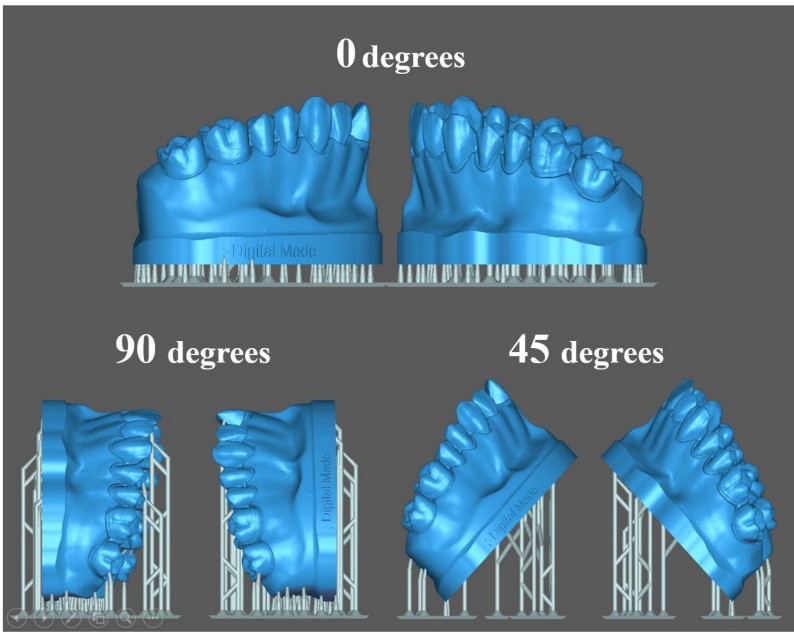

**Figure 2.** Printing orientation representations (0, 45, and 90 degrees).

All models were printed under controlled environmental conditions using the LCD printer (Phrozen, Sonic Mini 4K, Hsinchu, Taiwan), and the platform was calibrated before printing, according to the manufacturer's recommendations. Likewise, the printing parameters were predetermined following the resin manufacturer's suggestions (Phrozen Water Washable Dental Model Resin PHP-RS1000WWDM, Phrozen, Hsinchu, Taiwan). A layer thickness of 50 μ was established, with an exposure time of 35 s per layer set in the initial ones and 6 s in the rest. There was no need for holes for resin ejection, and they were printed with three angulations: 0, 45, and 90 degrees. For each build angle, ten models were printed. All models were printed with the same new-branded resin bottle and the same printer, using the same printing and post-processing protocol, except for the printing orientation, depending on the study group. The same operator trained in 3D printing carried out all the manufacturing.

After printing the models, they were post-processed. The models were placed in water for 15 min to remove excess impression material. Then, they were left for 15 min in the curing camera (Anycubic Wash&Cure 2.0 Shenzhen Anycubic Technology Co., Shenzhen, China). The support of all printed models was removed using a specific tool from the same manufacturer. All the models were stored in a dry area away from the light and without changes in temperature and humidity.

The implant analogs were manually inserted in the offset and screwed with a hexagonal 1.20 mm screwdriver and retained by friction. The scan body was manually placed and screwed. Scan powder (Vanishing Spray, SCANTIST 3D, Aztech Technologies, Singapore) was applied, and each analog's position was digitized by a laboratory scanner (Aidite A-IS Pro, Aidite Tech. Co., Qinhuangdao, China). The laboratory scanner was calibrated previously according to the manufacturer's recommendations. This manufacturer reports an accuracy of under 10 microns.

The accuracy analysis was performed using a specific reverse engineering software program (Geomagic Control X, v. 2020, 3D Systems, Rock Hill, SC, USA). The STL file from which all the models of the different groups to be evaluated were printed was the reference model with which the discrepancies of the experimental group files were measured. First, the reference STL was imported, and the mesh area corresponding to the scan body surface was selected. Subsequently, the experimental STL was imported. It was automatically superimposed by the best-fit alignment method, and the mesh area of the scan body surface was selected and compared with the reference one. A color map was obtained from the 3D comparison of the selected area (Figure 3). This process was carried out on each of the STLs of each of the 0-, 45-, and 90-degrees groups. The 3D root mean square (RMS) deviation values were obtained in millimeters (mm).

Precision was described as the RMS error variances per group standard deviation (SD). Trueness was defined as the average RMS error discrepancies between the reference file and the experimental model scans.

The statistical analyses of the accuracy of the implant analog position, according to the 3D printing angulation, were performed using a statistical software program (IBM SPSS Statistics for Windows, v.26.0; IBM Corp., Armonk, NY, USA). Descriptive statistics were performed by calculating each group's mean, median, and standard deviation. The Shapiro–Wilk test showed that the data were normally distributed. Therefore, the trueness and precision of the analog position in the 3D-printed models according to the printing orientation (0, 45, and 90 degrees) were analyzed using the parametric ANOVA test, establishing a significance level of $p < 0.05$, and subsequently the Bonferroni post hoc test.

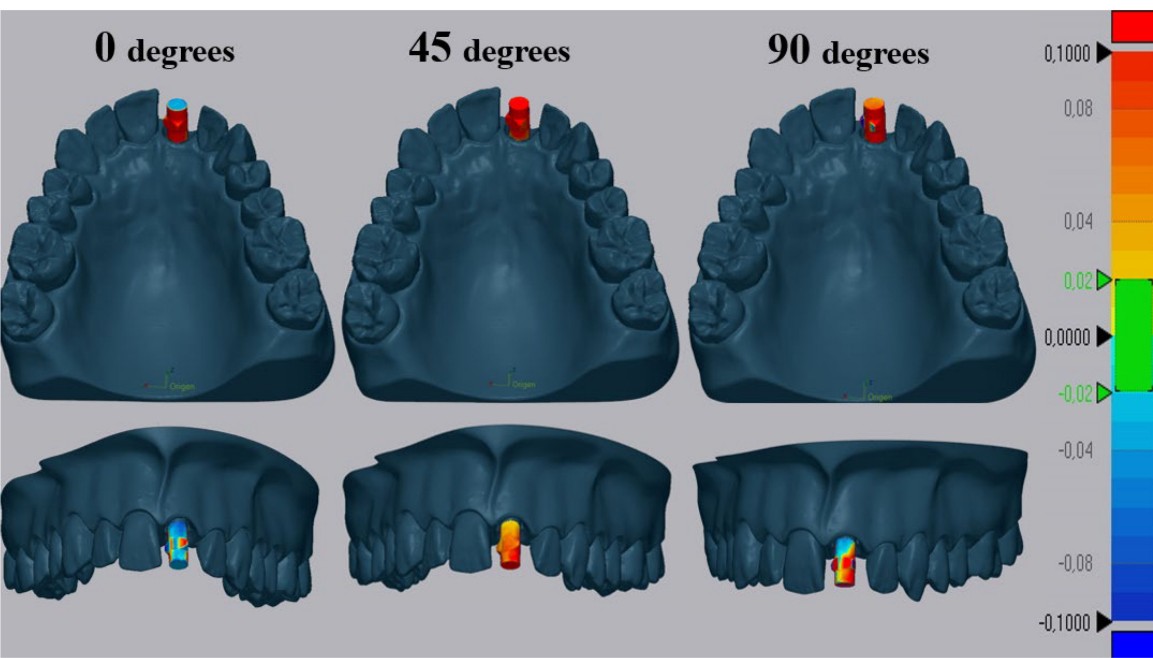

**Figure 3.** Representative images of color maps.

## 3. Results

A descriptive statistical analysis of the accuracy (trueness and precision) is presented in Table 1 and Figure 4, showing the mean, standard deviation, and minimum and maximum values for the three groups evaluated.

**Table 1.** Descriptive statistics for trueness (mm) and precision (mm).

|  |  | 0 Degrees | 45 Degrees | 90 Degrees |
|---|---|---|---|---|
| Mean ± SD | Trueness | 0.12 ± 0.02 | 0.10 ± 0.01 | 0.14 ± 0.03 |
|  | Precision | 0.10 ± 0.03 | 0.06 ± 0.01 | 0.10 ± 0.04 |
| Minimum value | Trueness | 0.09 | 0.09 | 0.11 |
|  | Precision | 0.06 | 0.04 | 0.07 |
| Maximum value | Trueness | 0.18 | 0.12 | 0.20 |
|  | Precision | 0.15 | 0.07 | 0.18 |

SD, Standard deviation.

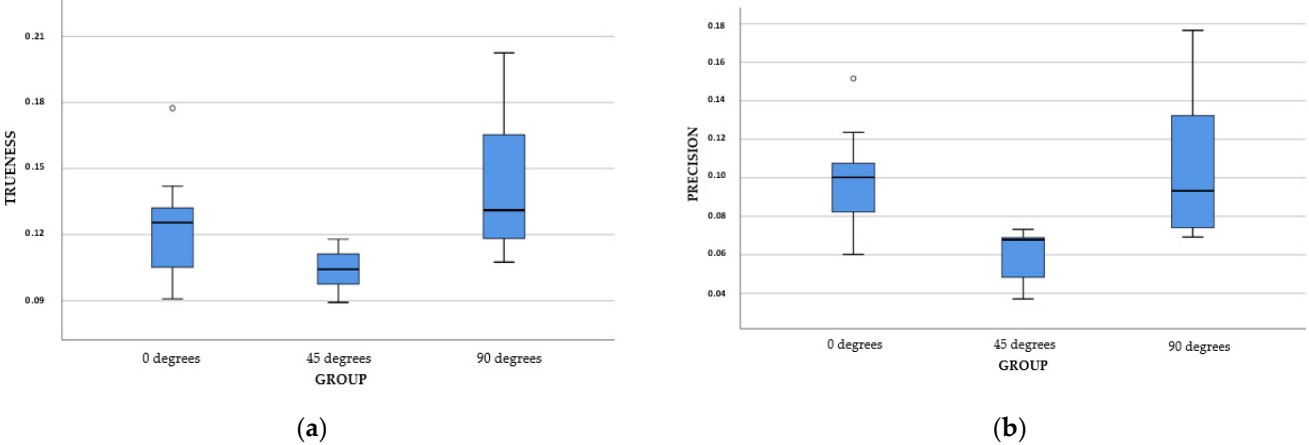

**Figure 4.** Representative box plot for accuracy (mm): (**a**) trueness; (**b**) precision.

The group with the highest trueness was the 45-degrees group (0.10 ± 0.01), followed by the 0-degrees group (0.12 ± 0.02) and the 90-degrees group (0.14 ± 0.03). The minimum

trueness value was 0.09 mm in the 0- and 45-degrees groups, followed by 0.11 mm in the 90-degrees group. Similarly, the maximum values found were 0.20 mm in the 90-degrees group, followed by 0.18 mm and 0.12 mm in the 0- and 45-degrees groups, respectively.

Concerning precision, the best values were found in the 45-degrees group ($0.06 \pm 0.01$), followed by the 0-degrees group ($0.10 \pm 0.03$) and the 90-degrees group ($0.10 \pm 0.04$). When evaluating the precision, the minimum value was 0.04 mm in the 45-degrees group, 0.06 mm in the 0-degrees group, and 0.07 mm in the 90-degrees group. Similarly, the maximum values found were 0.18 mm in the 90-degrees group, followed by 0.15 mm and 0.07 mm in the 0- and 45-degrees groups, respectively.

The ANOVA test showed statistically significant differences in trueness (RMS) among the groups ($p < 0.006$). The Bonferroni test showed that they were found between G2 and G3 (45 and 90-degrees of print orientation) ($p < 0.005$). No significant differences were found among the rest of the groups (Table 2).

**Table 2.** Comparison of trueness (mm) among the selected groups: 0-, 45-, and 90 degrees. Post hoc Bonferroni test.

| Group | Group | Mean Difference | Error Deviation | Sig. | 95% Confidence Interval Lower Limit | Upper Limit |
|---|---|---|---|---|---|---|
| **0** | 45 | 0.02 | 0.01 | 0.0198 | −0.07 | 0.05 |
| | 90 | −0.017 | 0.01 | 0.373 | −0.04 | 0.01 |
| **45** | 0 | −0.02 | 0.01 | 0.198 | −0.05 | 0.01 |
| | 90 | −0.04 | 0.01 | 0.005 | −0.06 | −0.01 |
| **90** | 0 | 0.02 | 0.01 | 0.373 | −0.01 | 0.04 |
| | 45 | 0.04 | 0.01 | 0.005 | −0.01 | 0.06 |

Precision was described as the RMS error variances of the standard deviations. The ANOVA test showed statistically significant differences between the groups ($p < 0.002$). In the post hoc Bonferroni test, the differences appeared between G1 and G2 (0 and 45-degrees of print orientation) ($p < 0.011$) and between G2 and G3 (45 and 90-degrees of print orientation) ($p < 0.003$) (Table 3).

**Table 3.** Comparison of precision (mm) among the selected groups: 0-, 45-, and 90 degrees. Post-hoc Bonferroni test.

| Group | Group | Mean Difference | Error Deviation | Sig. | 95% Confidence Interval Lower Limit | Upper Limit |
|---|---|---|---|---|---|---|
| **0** | 45 | 0.04 | 0.01 | 0.011 | 0.01 | 0.07 |
| | 90 | −0.01 | 0.01 | 1.000 | −0.04 | 0.02 |
| **45** | 0 | −0.04 | 0.01 | 0.011 | −0.07 | −0.01 |
| | 90 | −0.05 | 0.01 | 0.003 | −0.08 | −0.01 |
| **90** | 0 | 0.01 | 0.01 | 1.000 | −0.02 | 0.04 |
| | 45 | 0.05 | 0.01 | 0.003 | 0.01 | 0.08 |

## 4. Discussion

The accuracy of the 3D-printed implant models is a determinant factor in the passive fit of implant-supported prostheses. This study aimed to evaluate the accuracy of the implant analog position in 3D-printed models while varying the printing orientation (0, 45, and 90 degrees). The reported results revealed differences in the accuracy of the printed models depending on the printing angulation, so the null hypothesis was rejected.

Although the influence of the different impression parameters in the 3D-printed dental models has been previously studied, its impact on implant models remains unclear. The results of the present investigation indicate that the most accurate models are those manufactured at 45 degrees, followed by those printed at 0 and 90 degrees. In terms of

trueness, statistically significant differences were found between the 45- and 90-degrees groups ($p < 0.005$). No significant differences were found with the 0-degrees group. In terms of precision, there were statistically significant differences between the 45- and 0-degrees groups ($p < 0.011$) and between the 45- and 90-degrees groups ($p < 0.003$). No significant differences between the 0- and 90-degrees groups.

Print orientation has been reported as an influencing factor in the accuracy, material consumption, and mechanical properties of 3D models [30,31]. In terms of printing technology within additive manufacturing, DLP technology has been proven to be the most accurate technology, and it is known that the accuracy of DLP printing is further influenced by the optical specifications integrated with the system, such as lens quality, pixel size, DMD device, and platform resolution [21,32,33]. In addition, the storage of the printed models away from the light allowed no dimensional changes during the study, thus affecting the results [34].

Multiple factors affect the accuracy of the printed models with AM techniques, such as the building angle, printing orientation, layer thickness, the printer and the material used, the post-manufacturing shrinkage, and storage condition. Alshaibani et al. investigated the effect of these storage environmental conditions on the accuracy of 3D-printed models, determining that there were no statistically significant differences between environments with a temperature of $4 \pm 1$ or $20 \pm 2$ degrees Celsius [35]. Based on these results, the models of the present study were stored at a controlled temperature of $20 \pm 2$ degrees Celsius to prevent this factor from affecting the final accuracy of the models. In addition, Joda et al. also analyzed the impact of time on the accuracy of the models printed with AM technology, concluding that the dental models should not be used longer than 3 or 4 weeks after being manufactured [36]. Based on these results, the models in the present study were printed and stored until the complete sample was obtained, not exceeding two weeks between printing and digitalization, so this parameter would not interfere with the results.

The post-processing methods can also affect the performance of the printed samples, as the resins used suffer shrinkage and deformation [37–40]. In the present study, this process followed the manufacturer's instructions. It has also been noticed that this parameter can be reduced by increasing the photopolymerizing time or using microwave or UV radiation. It is also noteworthy that previous studies determined that building orientation also affects the mechanical properties of the printed specimens, affecting the passivity of the implant prostheses, establishing that models printed horizontally were the most affected. This conclusion agrees with the results obtained in the present study, where the best accuracy was reported in the 45-degrees group [13,41].

Previous studies have evaluated the impact of printing orientations. Rubayo et al. [30] used an SLA printer to print surgical splints, obtaining greater precision in the 0- and 45-degrees groups than in the 90-degrees group, so their results agreed with those obtained in the present study. However, Unkovskiy et al. evaluated the fabrication of complete prosthetic bases with DLP and SLA and found that the 90-degrees angulation provided greater trueness in both types of printers [31]. These differences may be explained by the printer models used, the geometry of the printed object, and the material used.

In tooth-supported crowns, the best accuracy was found at a 135-degree printing orientation [42]. This corresponds to an angulation of 45 degrees, coinciding with the most accurate impression angulation in the case of the present study. Other studies have also reported ranges between 30 and 90 degrees as the most accurate groups, depending on the printers used [43]. The evidence about the influence of the printing parameters in the accuracy of 3D implant printed models is limited.

In addition, it is worth noting that the 90-degree printing orientation allows for the manufacturing of more models simultaneously, which decreases the overall printing time of the specimens. An increase in print layers leads to an increase in printing time and possibly a higher probability of failure [44,45], so the 90-degree printing orientation also decreases resin waste by reducing the extent of the layers. In the present study, two models

per print run were used in all study groups until the sample size was complete so that this parameter did not influence the study's final results.

Discrepancies between groups may be due to multiple factors in addition to the printing orientation, such as the build starting points [46] or the scanner employed in digitizing the models [47,48]. Additionally, analog and scan body designs can affect the results. Further research focused on studying the influence of the different analog designs and materials would be recommendable.

The clinical application of the present study can be translated into a better fit of the implant-supported prostheses. There is currently no consensus among clinicians on the magnitude considered "unacceptable", as the data found in the literature varies up to 150 μm [39]. All values obtained were less than 140 μm, so all samples were clinically acceptable, with those printed at 45 degrees being the most accurate. Also, it is important to note that using 3D models in dentistry reduces the risk of accidentally damaging models and deleting medical data [49], making the clinical practice easier and communication with the laboratory and workflow faster.

The limitations of this in vitro study included the use of a 3D printer model and the resin type. Additionally, environmental conditions may influence the contraction of the 3D printing materials, affecting the mechanical properties and the position of the implant analog. Theoretically, it may lead to discrepancies that will impact the clinical level in the misfit of the prosthesis, causing mechanical and biological complications. It would be interesting to evaluate the different printing parameters in different clinical situations, varying the depth, angulation, number of implants, and distance between them. Furthermore, evaluating different printing resins and comparing various additive manufacturing technologies would be interesting.

## 5. Conclusions

Considering the limitations of this study, the following conclusions were drawn from the results obtained:

- The printing angulation parameter affects the accuracy of 3D-printed manufactured implant models.
- Implant models manufactured at 45 degrees of printing angulation are the most accurate, followed by those printed at 0 degrees and 90 degrees.

**Author Contributions:** Conceptualization, N.G., M.F. and R.C.; methodology, N.G., M.F. and R.C.; software, R.C. and J.L.A.-M.; validation, M.G.-P. and M.R.-L.; formal analysis M.G.-P., R.O., R.C. and C.G.-P.; investigation, N.G., M.F. and R.C.; resources, N.G., M.F. and J.L.A.-M.; data curation, R.O., R.C. and C.G.-P.; writing—original draft preparation, N.G. and R.C.; writing—review and editing, M.G.-P., M.R.-L. and R.C.; visualization, and M.R.-L.; supervision, R.C., M.F. and J.L.A.-M.; project administration, M.F. and R.C. All authors have read and agreed to the published version of the manuscript.

**Funding:** This research received no external funding.

**Institutional Review Board Statement:** Not applicable.

**Informed Consent Statement:** Not applicable.

**Data Availability Statement:** The data presented in this study are available on request from the corresponding author.

**Acknowledgments:** The authors would like to thank Digital Made laboratory for their assistance in this study, and the Dess group, for the material and technical support.

**Conflicts of Interest:** The authors declare no conflicts of interest.

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
