# Peer review of "Influence of Printing Angulation on the Accuracy (Trueness and Precision) of the Position of Implant Analogs in 3D Models: An In Vitro Pilot Study"

_applsci, doi:10.3390/app14072966_

Round 1

Reviewer 1 Report

Comments and Suggestions for Authors

This study investigates the effect of print orientation on the accuracy of 3D printed dental models, providing initial data on this specific parameter. While focusing solely on print angle addresses a particular research question, optimizing 3D print quality generally requires a multifaceted approach considering the many factors that can influence accuracy. The methodology utilizes common equipment and a limited sample size without clinical validation of printed models. Incorporating more advanced metrology tools, evaluating additional print parameters beyond orientation, using multiple implant positions, and validating with fabricated prostheses could expand the impact. The findings offer some useful insights on the effect of print angle, but further work would be needed to draw more robust conclusions to meaningfully enhance 3D printing processes in dentistry. Expansion of the methodology and research scope prior to publication could elicit richer, more broadly applicable results regarding accuracy of 3D printed dental models. However, this study represents a starting point in investigating the role of print orientation.

Author Response

Dear reviewer,

Thank you for accepting the review of this research study and for all your suggestions for improvement of the manuscript.

This study investigates the effect of print orientation on the accuracy of 3D printed dental models, providing initial data on this specific parameter. While focusing solely on print angle addresses a particular research question, optimizing 3D print quality generally requires a multifaceted approach considering the many factors that can influence accuracy. The methodology utilizes common equipment and a limited sample size without clinical validation of printed models. Incorporating more advanced metrology tools, evaluating additional print parameters beyond orientation, using multiple implant positions, and validating with fabricated prostheses could expand the impact. The findings offer some useful insights on the effect of print angle, but further work would be needed to draw more robust conclusions to meaningfully enhance 3D printing processes in dentistry. Expansion of the methodology and research scope prior to publication could elicit richer, more broadly applicable results regarding accuracy of 3D printed dental models. However, this study represents a starting point in investigating the role of print orientation.

Thank you for your suggestions and recommendations. As you rightly mention, it is true that there are multiple factors that influence 3D printing as explained in the manuscript. However, due to the rare literature regarding the subject, we have focused on evaluating the printing orientation as one of the factors that most affect 3D printing, standardizing and controlling the rest of the parameters, being a starting point for future research.

As for the sample size, it was determined based on the sample size used by different articles with similar objectives to the one presented, such as:

  • McCarty MC, Chen SJ, English JD, Kasper F. Effect of print orientation and duration of ultraviolet curing on the dimensional accuracy of a 3-dimensionally printed orthodontic clear aligner design. American Journal of Orthodontics and Dentofacial Orthopedics. 2020 Dec;158(6):889–97.

  • Ko J, Bloomstein RD, Briss D, et al. Effect of build angle and layer height on the accuracy of 3-dimensional printed dental models. Am J Orthod Dent Orthop. 2021;160:451e458.e2.

  • Charoenphol K, Peampring C. An In Vitro Study of Intaglio Surface, Periphery/Palatal Seal Area, and Primary Bearing Area Adaptation of 3D-Printed Denture Base

Manufactured in Various Build Angles. Munoz-Viveros CA, editor. International Journal

of Dentistry. 2022 Nov 17;2022:1–6.

  • Rubayo DD, Phasuk K, Vickery JM, Morton D, Lin WS. Influences of build angle on the

    accuracy, printing time, and material consumption of additively manufactured surgical

    templates. The Journal of Prosthetic Dentistry. 2021 Nov;126(5):658–63.

  • Jin SJ, Kim DY, Kim JH, Kim WC. Accuracy of Dental Replica Models Using Photopolymer Materials in Additive Manufacturing: In Vitro Three-Dimensional

    Evaluation. Journal of Prosthodontics. 2018 Jul 2;28(2):e557–62.

    Regarding the clinical validation of the models, we appreciate your proposal and we will take it into account for future research.

    Once again, thank you very much for your evaluation and suggestions. Best regards.

Reviewer 2 Report

Comments and Suggestions for Authors

1. The topic is current and important for modern medicine. In general, dental technicians adjust the position of the model relative to the plate or to the number of models without the presumption of accuracy and precision. The positioning of implants requires absolute accuracy because it is close to important anatomical structures – maxillary sinus, mandibular nerve, adjacent roots of natural teeth, etc.

2. The abstract is unstructured although properly written. Given that it is a study, it is more understandable to separate it into the same sections as the article - introductory sentence, material and methods, results and conclusion. The correction is minimal and will not inconvenience the authors.

3. The introduction is short, in terms of volume and authors. The topic is extensive and can be enriched.

4. Materials and methods are beautifully illustrated and explained.

5. The results are shown in tables and charts.

6. The discussion could be improved considering the large clinical application of the obtained results. I would personally apply what I learned in my practice.

7. The conclusion is short and clear. Please just add the result for the 0-degree angulation to the 45-and 90-degrees.

The article is of great scientific value for the laboratory work of modern dentistry. The corrections I described are minimal.

Author Response

Dear reviewer,

Thank you for accepting the review of this research study and for all your suggestions for improvement of the manuscript.

  1. The topic is current and important for modern In general, dental technicians adjust the position of the model relative to the plate or to the number of models without the presumption of accuracy and precision. The positioning of implants requires absolute accuracy because it is close to important anatomical structures – maxillary sinus, mandibular nerve, adjacent roots of natural teeth, etc.

We thank you for the recognition of the importance of the subject to be treated in the study, since one of the objectives is to obtain the most accurate printed models in order to obtain the most accurate implant-supported prostheses with the best fit.

  1. The abstract is unstructured although properly written. Given that it is a study, it is more understandable to separate it into the same sections as the article - introductory sentence, material and methods, results and conclusion. The correction is minimal and will not inconvenience the authors. 

Thank you for your suggestion. The abstract has been modified.

  1. The introduction is short, in terms of volume and The topic is extensive and can be enriched.

Thank you for your suggestion. The introduction has been rewritten.

  1. Materials and methods are beautifully illustrated and explained. 
  2. The results are shown in tables and charts. 
  3. The discussion could be improved considering the large clinical application of the obtained I would personally apply what I learned in my practice.

Thank you for your comments. The discussion has been modified.

  1. The conclusion is short and Please just add the result for the 0-degree angulation to the 45-and 90-degrees.

Thank you for your suggestion. The conclusion has been added.

Once again, thank you very much for your evaluation and suggestions. Best regards.

Reviewer 3 Report

Comments and Suggestions for Authors

The present article, even it does not represent an extraordinary novelty, it is welcome, the results being useful for the readership and a support to the dental practitioners.

 The manuscript is clear but some modifications should be done:

 -          The abstract has 260 words and should be a total of about 200 words maximum

-          Line 88 must be reformulated, some words are missing there. Not "the influence [...] is scarce, the knowledge about the influence is scarce.

-          the Figure 1b must be reversed, because the implant is in the position of the left central incisor, not in the right one

-          Almost half of the references are older than 5 years

Author Response

Dear reviewer,

Thank you for accepting the review of this research study and for all your suggestions for improvement of the manuscript.

- The abstract has 260 words and should be a total of about 200 words maximum. Thank you for your suggestion. The abstract has been modified.

The present article, even it does not represent an extraordinary novelty, it is welcome, the results being useful for the readership and a support to the dental practitioners.

The manuscript is clear but some modifications should be done:

- Line 88 must be reformulated, some words are missing there. Not "the influence [...] is scarce, the knowledge about the influence is scarce. The Figure 1b must be reversed, because the implant is in the position of the left central incisor, not in the right one.

Thank you for your comment. The errors has been modified.

- Almost half of the references are older than 5 years.

In relation to this reference, initially the bibliography search was based on articles from the last five years. When it became clear that the bibliography was scarce, the range of years was extended in order to increase the number of references and articles related to the topic to be addressed in this study. Even so, your consideration is appreciated and will be taken into account for the modifications of the manuscript.

Once again, thank you very much for your evaluation and suggestions. Best regards.

Reviewer 4 Report

Comments and Suggestions for Authors

1.In Introduction,please provide more specifics on the existing literature gap regarding orientation effects on implant model accuracy. Emphasize why your study is needed.

2.Compare your orientation findings with the existing studies on accuracy of tooth crowns and surgical guides to put the results into context.

Author Response

Dear reviewer,

Thank you for accepting the review of this research study and for all your suggestions for improvement of the manuscript.

1.     In Introduction, please provide more specifics on the existing literature gap regarding orientation effects on implant model accuracy. Emphasize why your study is needed.

Thank you for your suggestion. The introduction has been modified.

2.     Compare your orientation findings with the existing studies on accuracy of tooth crowns and surgical guides to put the results into context.

Thank you for your comments. This issue is resolved in the discussion section of the manuscript, specifically in paragraphs four and five.

Once again, thank you very much for your evaluation and suggestions. Best regards.
